# TRANSFERABLE UNSUPERVISED ROBUST REPRESENTATION LEARNING

## ABSTRACT

Robustness is an important, and yet, under-explored aspect of unsupervised representation learning, which has seen a lot of recent developments. In this work, we address this gap by developing a novel framework: Unsupervised Robust Representation Learning (URRL), which combines unsupervised representation learning's pretext task and robust supervised learning (*e.g.,* AugMix). Moreover, it is commonly assumed that there needs to be a trade-off between natural accuracy (on clean data) and robust accuracy (on corrupted data). We upend this view and show that URRL improves both the natural accuracy of unsupervised representation learning and its robustness to corruptions and adversarial noise. A further challenge is that the robustness of a representation might not be preserved in the transfer learning process after fine-tuning on downstream tasks. We develop transferable robustness by proposing a task-agnostic similarity regularization during the fine-tuning process. We show that this improves the robustness of the resulting model without the need for any adversarial training or further data augmentation during fine-tuning.

## 1 INTRODUCTION

Recently, there has been an increasing interest in unsupervised visual representation learning, where the goal is to learn effective representations of images without using human annotations (Bachman et al., 2019; Chen et al., 2020b; He et al., 2020; Misra & Maaten, 2020). In this work, we consider a relatively under-explored aspect: the *robustness* of these representations. Here, we use robustness to encompass the representation's resilience to common corruptions (Hendrycks & Dietterich, 2019), adversarial examples (Madry et al., 2018), and its ability to measure the uncertainty of its output in the face of such perturbations (Kumar et al., 2019).

More specifically, we develop new learning methods for *transferable robustness* of these representations, which improves and preserves the representation's robustness after fine-tuning on downstream tasks. While recent works have only evaluated the virtue of a representation by its accuracy on clean data after transfer learning and fine-tuning, we argue that transferable robustness should also be an integral part of a good representation. Recently, Rezaei & Liu (2020) have shown that downstream task models are vulnerable to adversarial attacks that are based solely on the pre-trained representation before fine-tuning. This highlights the importance of transferable robustness, without which downstream task models are vulnerable and lack generalization.

Moreover, it is commonly assumed that there needs to be a trade-off between natural accuracy (on clean data) and robust accuracy (on corrupted data) (Zhang et al., 2019). While there have been works showing that such a trade-off is not necessary for supervised learning (Yang et al., 2020), it is unclear how the principles they leverage is applicable to *unsupervised* representation learning.

**Summary of Contributions:** (1) We show that the accuracy-robustness trade-off is not necessary for unsupervised representation learning. (2) We develop a new representation learning framework: Unsupervised Robust Representation Learning (URRL), which uses a novel mix of representation learning pretext task and robust supervised learning (AugMix, Hendrycks et al. (2020b)). (3) In addition, we propose a task-agnostic similarity regularization that further improves the robustness of downstream tasks without the need for any adversarial training or additional data augmentation. (4) We introduce an evaluation framework for the transfer robustness of a representation, which includes its resilience to corruptions, adversarial robustness, and uncertainty calibration. (5) We show that

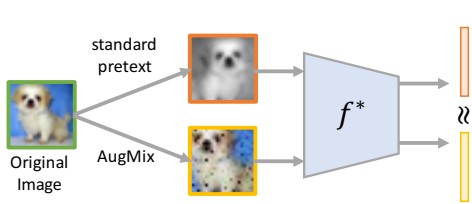 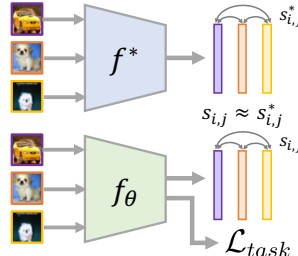

(a) Unsupervised Robust Representation Learning · (b) Similarity-based Regularization

Figure 1: (a) We show that by integrating AugMix with pretext tasks of state-of-the-art approaches. URRL leads to improvement of both clean accuracy and robustness and does not have the accuracy-robustness tradeoff. (b) We propose a task-agnostic similarity regularization, where the pairwise similarity $s_{i,j}$ of $f_\theta$ is regularized by that of the frozen representation $f^*$ during fine-tuning by $\mathcal{L}_{task}$. This encourages $f_\theta$ to capture $f^*$'s robustness without the need to train on corrupted data.

URRL improves both the clean accuracy and robustness of state-of-the-art representation learning under linear evaluation, and our full framework, URRL with Similarity Regularization (URRL-SR), further improves its robust accuracy by over 10% on 12 datasets.

The proposed URRL is shown in Figure 1(a). Our approach is motivated by the fact that in previous works, data augmentation has played an important role in both supervised learning robustness and unsupervised representation learning (Yun et al., 2019; Chen et al., 2020b). URRL randomly samples a pretext task of unsupervised representation learning and a robust data augmentation (AugMix, Hendrycks et al. (2020b)), and then optimizes them both under a contrastive learning framework. We show that URRL improves *both* the clean accuracy (on the original images without corruptions) and the robustness of the learned representation under the standard linear evaluation protocol.

Nevertheless, using the representation under linear evaluation limits the performance of transfer learning, and ultimately the usefulness of representation learning in downstream tasks. It is thus important to evaluate the representations' transferable robustness *after* fine-tuning the entire network to the downstream task. The biggest challenge is to preserve the inherited robustness of the representations: an aggressive fine-tuning schedule on a small dataset can significantly overwrite the robustness of the learned representations.

We address this challenge by proposing a task-agnostic similarity regularization that is applicable for the fine-tuning of the learned representation on any downstream task. We show that this improves the robustness of the fine-tuned model. The proposed regularization is shown in Figure 1(b). $f_\theta$ is initialized by the robust representation $f^*$ learned in Figure 1(a), and is now being fine-tuned by the downstream task loss $\mathcal{L}_{task}$. $f^*$ is robust because it is trained on augmentation of corrupted versions of an image.

We hypothesize that this form of robustness is not just about how $f^*$ represents an image individually, but also about how $f^*$ measures the similarities between images. This implies that we can preserve the robustness of $f^*$ by maintaining its similarity measures between images. We achieve this by regularizing the similarity matrix of a representation during fine-tuning to be similar to that of the original representation before fine-tuning. This allows the downstream task model to better utilize the robustness of the learned representation, while not capping the performance, which would be the case if the representation is fixed. Our full approach, URRL with similarity regularization (URRL-SR), further improves URRL's robust accuracy by over 10% on 12 datasets without any adversarial training or further data augmentation.

## 2 METHOD

In this work, we advocate the *transferable robustness* of unsupervised representation learning, which measures the robustness of a representation after fine-tuning on downstream tasks. The two challenges are: (i) It is commonly assumed that there is an accuracy-robustness trade-off, and the promotion of transferable robustness could potentially set back the performance gain from unsupervised

representation learning. (ii) the robustness of a representation could be changed drastically during the fine-tuning process, and thus make it challenging to preserve the robustness after fine-tuning the whole network. In Section 2.1, we address the first challenge by proposing URRL, which combines the advantages of pretext tasks from unsupervised representation learning with robust augmentations to improve not only the robustness but also the clean accuracy. In Section 2.2, we address the second challenge by proposing a task-agnostic similarity regularization that preserves the robustness of learned representations, while not capping the performance of transfer learning.

## 2.1 UNSUPERVISED ROBUST REPRESENTATION LEARNING

In this section, we discuss the proposed Unsupervised Robust Representation Learning (URRL), which combines the advantages of representation learning pretext task and robust augmentation. We observe that *data augmentation* plays a central role in the recent literature of *both* unsupervised representation learning and supervised learning robustness.

*Contrastive learning* has become a predominant design for representation learning (Chen et al., 2020b). Given an unlabeled dataset $\{x_i\}$ of images $x_i$, the goal is to learn the representation of each image $z_i = f_{\theta_1}(x_i)$ by optimizing the parameters $\theta_1$ of the encoding function. The role of data augmentation is to randomly create two correlated views of the same image. Let $\mathcal{T}_{rep}$ be a family of data augmentation. We can then sample two data augmentation functions $t_1, t_2 \sim \mathcal{T}_{rep}$, and apply them to an image $x_i$ to obtain a positive pair $x_q = t_1(x_i)$ and $x_{k+} = t_2(x_i)$. The goal is then to train the encoder $f_{\theta_1}$ such that it still encodes the two augmented images $x_q$ and $x_{k+}$ similarly despite the random augmentations to the images. In this case, the representation can learn to ignore information that is not task-relevant (introduced by the random augmentation), while still capturing the task-relevant information in the images. One can achieve this through the contrastive loss:

$$\mathcal{L}_{contra} = -\log \frac{\exp(z_q \cdot z_{k+}/\tau)}{\exp(z_q \cdot z_{k+}/\tau) + \sum_{k-} \exp(z_q \cdot z_{k-}/\tau)}, \tag{1}$$

where $\tau$ is the softmax temperature, and $z_{k-}$ are the representations of negative samples. The negative samples are often randomly augmented versions of images in the same batch $x_j, j \neq i$. The family of data augmentation $\mathcal{T}_{rep}$ plays a crucial role here as it would determine what the encoder $f_{\theta_1}$ learns to ignore and preserve in the representation of an image.

On the other hand, data augmentation improves the robustness of supervised learning by effectively expanding the training dataset without human annotation. Given a dataset $\{x_i, y_i\}$ of images $x_i$ and the corresponding labels $y_i$, one can apply data augmentation $\mathcal{T}_{rob}$ to obtain an augmented dataset $\{\mathcal{T}_{rob}(x_i), y_i\}$, where $\mathcal{T}_{rob}(x_i) = t(x_i), t \sim \mathcal{T}_{rob}$. $\mathcal{T}_{rob}$ improves the robustness of the model by indicating that the augmented image $\mathcal{T}_{rob}(x_i)$ still has the same label $y_i$ as the original image $x_i$. Recent works have shown that by mixing various types of perturbations, the learned classifier has the potential to be robust to novel perturbations that are not used in training (Hendrycks et al., 2020b). $\mathcal{T}_{rob}$ plays a crucial role in which types of perturbation the learned model would be robust to.

Given the importance of data augmentation in both of these aspects, a reasonable direction for unsupervised *robust* representation learning is to design a new family of data augmentation functions that can be used in Eq. (1) to learn robust representation without human annotation. We observe that the goal of $\mathcal{T}_{rep}$ and $\mathcal{T}_{rob}$ are quite similar at the high-level: In robustness literature, the aim of data augmentation is to increase the diversity of training data as much as possible without drifting off the data manifold (Hendrycks et al., 2020b). On the other hand, recent work hypothesizes that the best data augmentations for unsupervised representation learning are ones that "share the minimal information necessary to perform well at the downstream task" (Tian et al., 2020). If we take the downstream task to be classification and assume that the reason for not drifting off the data manifold is also optimizing for classification performance, then the aims of these two augmentations are actually quite similar: both are trying to preserve information that is enough to do well on classification, while increasing perturbation (and decreasing shared information) as much as possible. As we will show in the experiments, this is indeed the case: a representation learned with $\mathcal{T}_{rep}$ is already more robust than ones learned by supervised pre-training; At the same time, $\mathcal{T}_{rob}$ can already serve as an effective pretext task for unsupervised representation learning. It could seem that using $\mathcal{T}_{rob}$ as a pretext would directly lead to unsupervised representation learning with transferable robustness.

However, one missing piece of naively using $\mathcal{T}_{rob}$ as a pretext for representation learning is the augmentation that focuses on color distortion. As observed and shown by Chen et al. (2020b),

color distortion plays a critical role in the success of unsupervised representation learning because patches from the same image could share a similar color distribution, and without color distortion augmentation, the encoder could learn to exploit a shortcut to minimize Eq. (1) by focusing on the color histogram of an image. Drastic color distortion is often discouraged for $\mathcal{T}_{rob}$ because "histogram color augmentation can change the image's class" (Hendrycks et al., 2020b).

Therefore, we propose a straightforward solution to the problem by incorporating color histogram distortion into $\mathcal{T}_{rob}$. The proposed data augmentation family $\mathcal{T}_{urrl}$ for URRL does not sample the data augmentation function $t$ directly. Instead, we use a two-stage process:

$$\mathcal{T}_{urrl}(\boldsymbol{x}_i) = t(\boldsymbol{x}_i), \quad t \sim \mathcal{T}, \quad \mathcal{T} = \begin{cases} \mathcal{T}_{rob} & \text{with probability } p, \\ \mathcal{T}_{rep} & \text{with probability } 1-p, \end{cases} \tag{2}$$

where we first sample the family of data augmentation $\mathcal{T}$, and then sample the actual augmentation from $\mathcal{T}$. Here, the sampled $\mathcal{T}$ could be either $\mathcal{T}_{rob}$ or $\mathcal{T}_{rep}$. With the inclusion of $\mathcal{T}_{rep}$, $\mathcal{T}_{urrl}$ prevents the encoder $f$ from exploiting the color histogram. On the other hand, the use of $\mathcal{T}_{rob}$ in addition to $\mathcal{T}_{rep}$ further strengthens the robustness of the learned representation. We will show that this simple approach is able to outperform more sophisticated approaches to modify $\mathcal{T}_{rob}$. In this work, we use the pretext task of Chen et al. (2020d) as $\mathcal{T}_{rep}$ and AugMix (Hendrycks et al., 2020b) as $\mathcal{T}_{rob}$

## 2.2 SIMILARITY REGULARIZATION TOWARDS PRESERVED ROBUSTNESS

Now we have discussed the proposed Unsupervised Robust Representation Learning (URRL). The next question is how we can use and evaluate this learned representation. Once the representation encoder $f_{\theta_1}$ is learned, one either uses it to extract features of images for downstream tasks or perform *transfer learning* to further optimize and fine-tune $f_{\theta_1}$ on downstream tasks.

We consider a basic setting of transfer learning, where we have a learned representation $f_{\theta_1}$, and a function $g_{\theta_2}(\cdot)$ that can be applied to the extracted feature representation and transform it to a downstream task output. Given a dataset for the downstream task $\{\tilde{\boldsymbol{x}}_j, \tilde{\boldsymbol{y}}_j\}$, where $\tilde{\boldsymbol{x}}_j$ is the input image, and $\tilde{\boldsymbol{y}}_j$ is the target output, we assume that the downstream task would perform transfer learning by minimizing the following loss function:

$$\mathcal{L}_{task} = \ell(\tilde{\boldsymbol{y}}_j, \hat{\boldsymbol{y}}_j), \quad \hat{\boldsymbol{y}}_j = g_{\theta_2}(f_{\theta_1}(\tilde{\boldsymbol{x}}_j)), \tag{3}$$

where $\ell(\cdot, \cdot)$ is a loss function between the target $\tilde{\boldsymbol{y}}_j$ and the predicted output $\hat{\boldsymbol{y}}_j$. One can choose to minimize $\mathcal{L}_{task}$ by optimizing only $\theta_2$ or both $\theta_1$ and $\theta_2$. The downside of the first option is that the performance might be limited because $f_{\theta_1}$ could potentially ignore information that is useful for the downstream task. A more common approach is thus to also optimize $f_{\theta_1}$ with $\mathcal{L}_{task}$. This has shown to be a successful approach for a variety of downstream tasks, whether $f_{\theta_1}$ is learned with supervised pre-training or unsupervised representation learning.

Nevertheless, one missing aspect is the robustness of the resulting model after optimizing $f_{\theta_1}$ and $g_{\theta_2}$ with $\mathcal{L}_{task}$. As the robustness is not part of the objective, $f_{\theta_1}$'s robustness could be drastically changed after the optimization, and there is no guarantee that the resulting model would still be robust even if we start with a $f_{\theta_1}$ that is robust. One option is to design robust optimization methods for the downstream task, for example, by adversarial training (Madry et al., 2018) or even by using robust augmentations again on the downstream task. However, this is more time-consuming (Shafahi et al., 2019) and might even need to be task-specific (Zhang & Wang, 2019).

To this end, we propose a simple yet effective task-agnostic regularization to supplement the transfer learning loss in Eq. (1) and preserve the robustness of the learned feature representation. The proposed regularization does not significantly increase the training computation and can be easily combined with any form of $\mathcal{L}_{task}$. This allows us to better preserve the robustness of learned representations during fine-tuning and improve the robustness of downstream task models.

The robustness of a $f_{\theta_1}$ comes from its ability to represent an image similarly despite the perturbations. $\mathcal{L}_{contra}$ in Eq. (1) encourages $f_{\theta_1}$ to represent augmented (and potentially corrupted) versions of an image closer than that of other images. This property is beneficial to downstream task robustness because if $f_{\theta_1}(\tilde{\boldsymbol{x}}_j)$ is less affected by a perturbation, then it is more likely that $\hat{\boldsymbol{y}}_j = g_{\theta_2}(f_{\theta_1}(\tilde{\boldsymbol{x}}_j))$ can be robust to the perturbation.

We hypothesize that this form of robustness is not just about how $f_{\theta_1}$ represents an image individually, but also about how $f_{\theta_1}$ measures the similarities between images. This implies that we can

Table 1: Linear evaluation results of transferable robustness. URRL combines the advantages of AugMix and MoCo v2 to improve on all of the proposed linear evaluation metrics.

|  | MoCo v2 | AugMix | AugMix (AllOp) | AugMix (MoCo) | URRL |
|---|---|---|---|---|---|
| ImageNet top-1 | 67.59 | 61.45 | 61.97 | 64.41 | **68.47** |
| ImageNet-C mCE | 88.21 | 94.47 | 92.34 | 90.18 | **84.41** |
| Calibration Error | 15.30 | 16.17 | 15.44 | 15.31 | **14.64** |

preserve the robustness of $f_{\theta_1}$ by maintaining its similarity measures between images. Therefore, we propose a similarity matrix regularization to preserve the robustness of $f_{\theta_1}$ during transfer learning:

$$\mathcal{L}_{sim} = ||\boldsymbol{S} - \boldsymbol{S}^*||, \quad S_{i,j} = f_{\theta_1}(\tilde{\boldsymbol{x}}_i) \cdot f_{\theta_1}(\tilde{\boldsymbol{x}}_j), \quad S^*_{i,j} = f^*(\tilde{\boldsymbol{x}}_i) \cdot f^*(\tilde{\boldsymbol{x}}_j), \quad (4)$$

where $f^*(\cdot)$ is the frozen version of $f_{\theta_1}$ before fine-tuning. There are two advantages of this regularization: First, we do not significantly increase the computational requirement, since $f_{\theta_1}(\tilde{\boldsymbol{x}}_i)$ is already needed, and $f^*(\tilde{\boldsymbol{x}}_i)$ is just a single forward pass of the dataset. Second, it is task agnostic: only the representation encoder $f_{\theta_1}$ and input data $\tilde{\boldsymbol{x}}_i$ are needed. The regularization is not affected by any of the task-specific terms in Eq. (3) ($\tilde{\boldsymbol{y}}_j$, $\ell(\cdot, \cdot)$, $g_{\theta_2}(\cdot)$). Ideally one would want to make sure $f_{\theta_1}$ still preserves the similarities on the augmented data used to pre-train the representation. However, we observe that simply using the downstream task input data is sufficient, while being much more efficient. In practice, we construct $\boldsymbol{S}$ and $\boldsymbol{S}^*$ using the images in the same batch.

## 3 EXPERIMENTS

We have discussed the proposed URRL, and how we can preserve its robustness during transfer learning using similarity regularization. Now we aim to evaluate both its performance and transferable robustness. We evaluate the fine-tuned model from three aspects: (i) resilience to common corruptions, (ii) adversarial examples, and (iii) its ability to measure the uncertainty of its output in the face of such perturbations. This provides a holistic view of a representation's transferable robustness.

### 3.1 ROBUSTNESS WITH LINEAR EVALUATION PROTOCOL

**Experimental setup.** Linear evaluation has been the most popular way to evaluate unsupervised representations learning. In this setting, the representation is not changed, and a linear classifier is trained on top of the representation with the full ImageNet (Deng et al., 2009). Once trained, this classifier is evaluated on the clean ImageNet without any perturbation. We further evaluate this classifier trained on clean data directly on the perturbed dataset ImageNet-C, in which images from ImageNet are perturbed by 15 types of algorithmically generated corruptions from noise, blur, weather, and digital categories (Hendrycks & Dietterich, 2019).

**Metrics.** On the clean ImageNet, we report the top-1 accuracy following previous works. For ImageNet-C, the standard metric is mean Corruption Error (mCE), where we report the error rate normalized by the error rate of AlexNet, and average across the 15 types of corruption. The reason for this normalization is that different types of corruption have different difficulties, and we do not want the evaluation to be skewed towards a small set of corruptions. In addition to mCE, we further evaluate the RMS Calibration Error to assess the models' uncertainty estimates. The calibration error measures how well the model's confidence in its prediction matches the actual accuracy. For a calibrated model, if a prediction has 70% confidence, then it should have 70% chance of being correct. We follow previous work and use a discrete approximation of this metric, where each bin has 100 predictions (Hendrycks et al., 2020b).

**Baselines.** We compare with the following representation learning approaches:

- *MoCo v2* (Chen et al., 2020d). We select MoCo v2 as the baseline and build all of the following approaches on top of its publicly available official codebase. We select MoCo v2 due to its state-of-the-art performance and lower requirement for batch size. Our approach and baselines can easily adapt and be built on other unsupervised representation learning approaches. In Appendix B, we

Table 2: Transferable adversarial robustness results. The highest accuracy is marked with bold. Our full approach URRL-SR has the highest robust accuracy for most of the datasets among representation learning approaches, while losing much less clean accuracy than adversarial training. This is achieved without adversarial training and further data augmentation.

| | Aircraft | Birdsnap | CIFAR10 | CIFAR100 | Caltech101 | Caltech256 | Cars | DTD | Flowers | Food | Pets | SUN397 | Average |
|---|---|---|---|---|---|---|---|---|---|---|---|---|---|
| **robust accuracy**: | | | | | | | | | | | | | |
| ImageNet | 1.65 | 2.01 | 16.18 | 11.83 | 28.51 | 19.60 | 3.07 | 21.76 | 26.85 | 7.60 | 8.09 | 2.92 | 12.51 |
| MoCo v2 | 8.22 | 5.73 | 39.37 | 15.49 | 56.86 | 26.40 | 24.19 | 36.76 | 51.20 | 10.44 | 17.66 | 9.03 | 25.11 |
| URRL | 10.17 | 6.17 | 39.85 | 14.95 | 57.61 | 25.75 | 22.83 | 35.80 | 54.56 | 10.35 | 17.55 | 8.87 | 25.37 |
| URRL-L2SP | 16.47 | 8.11 | 40.68 | 17.84 | **59.15** | 28.42 | 27.84 | 37.23 | **55.08** | 15.86 | 21.72 | 8.59 | 28.08 |
| MoCo-SR | 25.71 | 11.88 | 69.48 | 16.95 | 55.94 | 33.55 | 42.76 | **40.64** | 49.59 | 33.57 | 40.86 | **14.59** | 36.29 |
| URRL-SR | **26.55** | **11.93** | **70.44** | **19.55** | 56.81 | **33.63** | **42.94** | 39.89 | 49.88 | **34.28** | **40.94** | 13.76 | **36.72** |
| Adv. Training | 45.84 | 42.49 | 87.93 | 64.44 | 39.06 | 44.80 | 48.22 | 25.80 | 36.25 | 66.43 | 19.27 | 34.46 | 46.25 |
| **clean accuracy**: | | | | | | | | | | | | | |
| ImageNet | 83.17 | **72.49** | 97.27 | **84.73** | **93.34** | **82.10** | 87.71 | **74.84** | 95.15 | 87.37 | **93.73** | 62.41 | **84.53** |
| MoCo v2 | **87.69** | 70.76 | **97.54** | 83.04 | 85.09 | 70.77 | 89.64 | 69.20 | 94.77 | 87.69 | 86.92 | 57.04 | 81.68 |
| URRL | 87.04 | 70.91 | 97.34 | 83.34 | 85.60 | 70.64 | 89.91 | 68.72 | 95.09 | 87.91 | 87.38 | 57.38 | 81.77 |
| URRL-L2SP | 87.37 | 71.24 | 97.37 | 83.55 | 85.34 | 72.48 | **89.95** | 70.85 | 95.76 | **88.06** | 88.74 | 57.75 | 82.37 |
| MoCo-SR | 77.73 | 60.66 | 97.24 | 80.49 | 86.49 | 67.98 | 79.31 | 68.14 | 95.17 | 86.19 | 88.83 | 46.61 | 77.85 |
| URRL-SR | 78.55 | 60.92 | 97.10 | 80.60 | 88.38 | 67.61 | 80.66 | 68.62 | **95.87** | 86.74 | 88.83 | 46.72 | 78.38 |
| Adv. Training | 57.52 | 56.86 | 92.94 | 74.30 | 42.80 | 51.97 | 58.97 | 31.33 | 40.66 | 79.63 | 23.06 | 44.31 | 54.53 |

show how URRL can be used to improve other unsupervised learning approaches. Results in the main paper would focus on the relative improvement from MoCo v2 for a fair comparison.

*- AugMix* (Hendrycks et al., 2020b). We select AugMix as the family of robust data augmentation due to its simplicity and effectiveness. Here, we simply replace the pretext task of MoCo v2 with AugMix for unsupervised representation learning using Eq. (1).

*- AugMix (AllOp).* In the original AugMix paper, augmentation types `contrast`, `color`, `brightness`, and `sharpness`, are excluded as they involve more drastic color distortion and may overlap with ImageNet-C corruptions. However, these types of corruption are already used in the pretext task of MoCo v2. Therefore we add these operations back for a fair comparison.

*- AugMix (MoCo).* We can further strengthen the color distortion of AugMix by including the data augmentations used in MoCo v2 *within* the random convex combinations of AugMix. For example, MoCo v2 randomly makes the cropped patch grayscale as data augmentation. However, in AugMix, this only happens in the extreme case of `color` augmentation.

*- URRL.* Finally, our URRL in Eq. (2) introduces color distortion to robust augmentation by hierarchically and randomly selecting the family of data augmentation to use. We use the pretext task of MoCo v2 for $\mathcal{T}_{rep}$, and AugMix (AllOP) for $\mathcal{T}_{rob}$.

**Implementation Details.** We build on the official repo of MoCo v2, and use the exact same hyperparameters for all methods (200 epochs with learning rate 0.3). We use $p = 0.5$ in Eq. (2).

**Results.** The results are shown in Table 1. Directly using AugMix as a pretext task is not ideal, however, the performance is already comparable to competitive approaches in the same setup (He et al., 2020; Chen et al., 2020b). By gradually increasing the severity of color distortion augmentations, AugMix (AllOp) and AugMix (MoCo) continue to improve both the clean accuracy (ImageNet top-1) and mCE. URRL performs the best on all three metrics by combining the data augmentation from MoCo v2 and AugMix. The randomly selected family of data augmentation could involve drastic color distortion (when $\mathcal{T}_{rep}$ is selected), and thus the model cannot take the shortcut by looking at the color histogram. On the other hand, the corruption and perturbations in AugMix complement well with the pretext used in MoCo v2, and even improves the ImageNet top-1 accuracy.

## 3.2 EVALUATING TRANSFERABLE ADVERSARIAL ROBUSTNESS

While it is helpful to evaluate representations under linear evaluation, the performance is limited when we fine-tune the representations for other datasets. In this section, we evaluate the performance and robustness of representations after transfer learning and fine-tuning the whole model.

Table 3: RMS calibration errors on CIFAR-10-C and CIFAR-100-C. Similarity regularization significantly improves models' calibration.

|  | MoCo v2 | URRL | MoCo-SR | URRL-SR |
|---|---|---|---|---|
| CIFAR-10-C | 16.96 | 17.90 | 9.92 | **8.95** |
| CIFAR-100-C | 19.05 | 16.39 | 12.60 | **11.25** |

**Experimental Setup.** We consider 12 common datasets used for evaluating transfer learning of representation learning (Salman et al., 2020). If not noted otherwise, the representation is trained with the transfer learning loss in Eq. (3), where $g_{\theta_2}$ is a linear classifier on top of the representation, and $\ell(\cdot, \cdot)$ is the cross entropy loss for classification.

**Metrics.** We report the top-1 clean accuracy on these datasets. In addition, we take the models and report the robust accuracy from adversarial evaluation. We follow the settings of Ilyas et al. (2019).

**Methods for Comparison.** We compare MoCo v2 and the proposed URRL with and without similarity regularization. We refer to the similarity-regularized versions as MoCo-SR and URRL-SR. We use $L^2$-SP as the baseline for transfer learning regularization, which directly regularizes the fine-tuned weights to be similar to that of the pre-trained weights (Li et al., 2018). Our approach with $L^2$-SP is referred to as URRL-L2SP. We also evaluate ImageNet pre-trained weights to compare the transferable robustness of supervised and unsupervised representations. We show results of adversarial training on each of the 12 datasets as a reference for robust accuracy, but it is important to note that it is not a fair comparison as it performs much heavier adversarial optimization for downstream tasks.

**Implementation Details.** We use the ResNet-50 backbone as in Section 3.1, which has a $7 \times 7$ conv1 instead of a $3 \times 3$ conv1 often modified for CIFAR-10. We resize the images to $256 \times 256$ to feed into our network. We select the best learning rate and the weight for $\mathcal{L}_{sim}$ on a validation set, and for the rest of the parameters, we use the same as Salman et al. (2020). We use the code and parameters of Ilyas et al. (2019) for adversarial training and evaluation ($\ell_2$ attack with $\epsilon = 0.5$).

**Results.** The results for clean accuracy and robust accuracy are shown in Table 2. ImageNet pre-trained weights have the best clean accuracy in our experiment. This is consistent with results in previous works that use the same ResNet-50 backbone and computing budgets (200 epochs) as our experimental setup (He et al., 2020; Chen et al., 2020b). However, ImageNet pre-trained representation has the lowest robust accuracy by a large margin. Unsupervised representation learning (MoCo v2) is already more robust than ImageNet pre-trained weights. URRL is able to further improve both the clean and robust accuracy, and regularizing the weights to be similar to that of the pre-trained weights is also effective (URRL-L2SP). Nevertheless, the proposed similarity regularization significantly improves the robust accuracy by over 10% on average for both MoCo-SR and our URRL-SR, with a much smaller loss of clean accuracy compared to that of adversarial training. URRL-SR has the highest robust accuracy for most of the datasets among representation learning approaches, while losing much less clean accuracy than adversarial training. This shows that both the proposed URRL and similarity regularization are able to improve the transferable robustness without much overhead. In particular, the proposed similarity regularization is also better for robustness compared to $L^2$-SP because similarity measure loss directly regularizes the robustness induced by contrastive learning on augmented data during training.

### 3.3 UNCERTAINTY CALIBRATION OF TRANSFER LEARNING

We further evaluate the uncertainty calibration of transfer learning. Improving the uncertainty calibration of downstream task models is an important aspect of transferable robustness. If the learned representation can improve calibration of downstream tasks, then even if the downstream task model cannot handle all kinds of new inputs outside its training distribution, it at least knows when it is likely to be mistaken. This alleviates the potential impact of unforeseen inputs.

**Experimental Setup.** We use CIFAR-10-C and CIFAR-100-C for this experiment, where perturbations that are similar to ImageNet-C are applied to CIFAR-10 and CIFAR-100 instead of ImageNet (Hendrycks & Dietterich, 2019). We fine-tune models on CIFAR-10 and CIFAR-100 and

evaluate on CIFAR-10-C and CIFAR-100-C, respectively. We evaluate the RMS Calibration Error following the previous work of Hendrycks et al. (2020b), where each bin has 100 predictions for approximation.

**Results.** The RMS calibration errors are shown in Table 3. The proposed similarity regularization significantly improves the calibration error on both datasets (half the error for URRL-SR compared to URRL on CIFAR-10-C). URRL-SR further improves calibration error compared to MoCo-SR. This shows the importance of similarity regularization in combination with the proposed URRL for transfer learning robustness.

## 4 RELATED WORK

**Unsupervised representation learning.** There have been significant recent advances in unsupervised representation learning. Contrastive learning (Gutmann & Hyvärinen, 2010) has become a predominant design leading to state-of-the-art performance (Oord et al., 2018; Bachman et al., 2019; Chen et al., 2020c;b; He et al., 2020; Misra & Maaten, 2020; Tian et al., 2020; Chen et al., 2020d). Latest methods even outperform supervised pre-training on several transfer learning tasks (Caron et al., 2020). Most existing works benchmark representations by directly fine-tuning the pre-trained models on downstream tasks. However, as observed by He et al. (2019), ImageNet pre-training does not necessarily improve performance on large-scale downstream tasks under such settings. We believe part of the reason lies in the catastrophic forgetting, especially when the scale of a downstream task is large. The proposed similarity regularization casts new insights towards addressing this issue.

**Robustness from perturbed inputs.** Besides playing an important role in unsupervised representation learning, training with perturbed/corrupted inputs has been widely shown to render stronger robustness against adversarial attacks (Eghbal-zadeh et al., 2020; Madry et al., 2018) and corruption (Hendrycks et al., 2020b;a), as well as improved network attention towards better task performance (Ghiasi et al., 2018; Singh & Lee, 2017; Yun et al., 2019; Zhong et al., 2020) . Similar approaches (Cubuk et al., 2020) have also been widely adopted in unsupervised domain adaptation (French et al., 2018) and semi-supervised learning (Berthelot et al., 2019b;a; Laine & Aila, 2017; Sohn et al., 2020; Xie et al., 2020). We develop new methods to improve the transferable robustness of unsupervised representation learning, which is an important yet under-explored area.

**Robustness and representation learning.** In this work, we show that combining a standard pretext task with AugMix improves both the clean accuracy and robustness of unsupervised representation learning. This is consistent with recent findings on the benefit of having better representation for *supervised* adversarial robustness. It has been shown that self-supervised learning (Chen et al., 2020a; Hendrycks et al., 2019b; Jiang et al., 2020; Kim et al., 2020), supervised pre-training (Hendrycks et al., 2019a), unlabeled data (Alayrac et al., 2019; Carmon et al., 2019), and multi-task learning (Mao et al., 2020) are beneficial to adversarial robustness when jointly optimized with adversarial training. However, the main goal of these approaches is to complement *supervised* adversarial training, whether the approach itself is supervised or unsupervised. The robustness of unsupervised representation learning is rather under-explored. Recent work even shows that self-supervised model lacks noise robustness (Geirhos et al., 2020). Our goal is to address this challenge and improve the transferable robustness of representations without adversarial training and further data augmentation. The most related work to ours are Alayrac et al. (2019) and Kim et al. (2020), which aim to improve adversarial robustness without labeled data. However, we still see an accuracy-robustness trade-off. Our work shows that the trade-off is not necessary for unsupervised representation learning, and covers a wider range of robustness, such as common corruptions and uncertainty calibration.

**Transfer learning robustness.** Our emphasis on the transferable robustness is closely related to the works on the robustness of transfer learning. Rezaei & Liu (2020) have shown that downstream task models are vulnerable to attacks based on the pre-trained representation. Chin et al. (2020) proposes a noisy feature distillation as a defense for the aforementioned attack. Shafahi et al. (2020) also shows that a learning without forgetting (LwF) (Li & Hoiem, 2017) based loss from an adversarially trained source model can improve adversarial robustness of the downstream task model. Our goal of learning more robust representation without supervision is mutually beneficial to these efforts by serving as a better starting point for robust transfer learning. Our similarity regularization has a similar goal with the LwF loss, but is less restrictive (only pairwise relationship constraint). Since these

regularizations do not require test-time labels, they can also be applied to test-time optimization (Sun et al., 2020) to regularization the self-supervised learning loss in the target/test domain.

## 5 CONCLUSIONS

We present Unsupervised Robust Representation Learning (URRL), a new framework that combines the advantages of unsupervised representation learning and supervised learning robustness. This upends the commonly assumed accuracy-robustness trade-off in the case of unsupervised representation learning. Moreover, we show that the proposed similarity regularization can preserve URRL's robustness during fine-tuning. Our full model URRL-SR significantly improves the downstream task robustness without adversarial training and further data augmentation. We believe an impactful future direction is how we can learn a representation that is still robust after fine-tuning even without any regularization. While our regularization is task-agnostic, it is still desirable that one can straightforwardly improve the robustness of fine-tuned models without any modification.

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

# A $L^2$-SP HYPERPARAMETER ANALYSIS

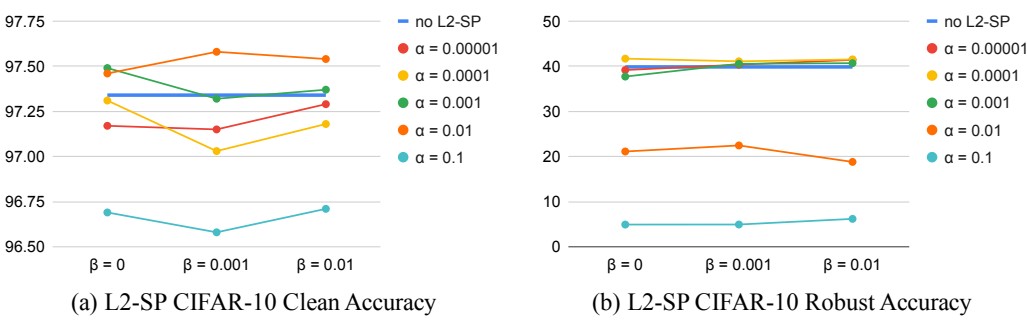

(a) L2-SP CIFAR-10 Clean Accuracy     (b) L2-SP CIFAR-10 Robust Accuracy

Figure 2: $L^2$-SP results on CIFAR-10.

Figure 2 shows the $L^2$-SP results on CIFAR-10, where we conduct the same hyperparameters analysis as Li et al. (2018). We can see that their best performing hyperparameters ($\alpha = 0.1, \beta = 0.01$) actually hurt both the clean accuracy and robust accuracy in our case. This is due to two key differences of our work compared to theirs: First, our focus is on *unsupervised representation learning*. Compared to the representations learned from supervised learning on large-scale datasets like ImageNet and Places365, our unsupervised representations typically need further updates in order to do well on downstream classification tasks. In this case, $L^2$ regularization is not enough by itself to improve the performance. Second, *robustness* is not the goal of Li et al. (2018), and $L^2$-SP is not designed to improve the robustness of transfer learning. In contrast, the primary goal of our work is to improve the transferable robustness of learned representations, and the design of our similarity measure loss is motivated by preserving the robustness of learned representations during fine-tuning. As discussed in Section 2.2, the robustness of a feature representation comes from how it measures the similarities between images, and the proposed approach explicitly regularizes it during fine-tuning. Our results in Table 2 also show the importance of the proposed approach.

# B GENERALITY OF OUR AUGMENTATION SCHEME

Table 4: Results on applying URRL to multiple unsupervised representation learning approaches.

|  | ImageNet top-1 | ImageNet-C mCE | Calibration Error |
|---|---|---|---|
| InstDis | 59.50 | 104.93 | 19.86 |
| InstDis w/ URRL | **60.42** | **101.64** | **18.30** |
| PIRL | 61.70 | 103.15 | 18.81 |
| PIRL w/ URRL | **62.47** | **99.74** | **18.19** |

We conduct the experiments in Section 3.1 on other unsupervised representation learning approaches, including InstDis (Wu et al., 2018) and PIRL (Misra & Maaten, 2020) to show the generality of our approach. Results are shown in Table 4. InstDis w/ URRL means that we use InstDis as $\mathcal{T}_{rep}$ and AugMix (AllOp) as $\mathcal{T}_{rob}$ in Eq. (2). Similarly for PIRL and PRIL w/ URRL. URRL improves both of these methods, which supports the generality of our approach. We hope this facilitates its broad adoption in self-supervised/unsupervised learning

