# OpenReview forum: "Transferable Unsupervised Robust Representation Learning"
_ICLR.cc/2021/Conference — Reject_

### Official Review · AnonReviewer1 · 2020-10-27
**A Work on Making Self-Supervised Learning Endow Robustness**

**Rating:** 7
**Confidence:** 5

**Review:**

This paper uses a different data augmentation (AugMix) scheme to improve self-supervised representation learning. It improves accuracy and (corruption and adversarial) robustness by a sufficiently interesting amount. The paper's presentation is clear, but the paper could be more thorough. Since the technique is simple and general, it could easily be broadly applicable to the burgeoning area of self-supervised learning.

Other pros:
_On the surprising similarities between supervised and self-supervised models_ shows that most techniques for self-supervised learning don't improve robustness at all, so self-supervised learning produces representations that have limited use downstream.
This work counteracts this limitation by also assessing the robustness of self-supervised representations, and the technique they propose improves robustness. They improve robustness with data augmentation, a volatile ingredient in self-supervised learning. SimCLR reminds us that the choice of image modifications is crucial and must be carefully selected, so it was not obvious a priori that their technique should help.

Other cons:
The paper would be stronger if it showed their augmentation scheme helping with more than MoCo. That would demonstrate generality. I'll boost my score if it improves another self-supervised learning technique.
The paper could show results with another robustness benchmark, such as ImageNet-R. This would make a stronger case for robustness. This would be quick to add.
URRL-SR uses embedding distances, but this seems like logit pairing. Logit pairing can lead to an overestimate of adversarial robustness. Please include results with an adversarial attack with restarts. Also state whether you're using l2 or l_infty attacks, rather than having refer to Ilyas et al.
The paper has one only figure, and Table 2's formatting is slightly unintuitive.

Update: I am happy with the changes and am keeping my score.

---

> ### Author Response · Authors · 2020-11-24
> **Our Response to Reviewer 1**
>
> We thank the reviewer for the helpful discussions and several excellent suggestions to strengthen our paper. We would also like to thank the reviewer for a nice summary of our results, which we will use to revise our paper. We have included additional results in Table 4 of Appendix B, where we show the generality of our approach. Our approach (URRL) improves two other unsupervised learning approaches ([InstDis](https://openaccess.thecvf.com/content_cvpr_2018/CameraReady/0801.pdf), [PIRL](https://openaccess.thecvf.com/content_CVPR_2020/papers/Misra_Self-Supervised_Learning_of_Pretext-Invariant_Representations_CVPR_2020_paper.pdf)). We believe this result supports the generality of our approach. We hope its simplicity could be viewed as a virtue that facilitates its broad adoption in self-supervised/unsupervised learning. We have also updated our paper with discussion and more details of our adversarial training/evaluation. We are continuing to revise our paper to incorporate further experiments suggested by the reviewer.

---

> > ### Comment · AnonReviewer1 · 2020-11-24
> > **Appendix B?**
> >
> > I do not see any appendix for the current PDF on OpenReview.

---

> > > ### Author Response · Authors · 2020-11-24
> > > **Updated PDF**
> > >
> > > Just updated the pdf. Could you check the pdf again? Thanks! Was posting other responses.

---

### Official Review · AnonReviewer2 · 2020-10-28
**Minimal and insufficiently motivated changes to existing methods**

**Rating:** 5
**Confidence:** 4

**Review:**

## Paper summary
This paper proposes *Unsupervised Robust Representation Learning* (URRL), a framework that combines several data augmentation schemes and a similarity-based loss. The goal is to improve the robustness of visual representations to image perturbations. A further goal is to maintain the robustness properties of pre-trained representations after fine-tuning the network to downstream tasks.

The proposed method consist of two components: A scheme for sampling combinations of image augmentations, and a loss that is applied during transfer learning and penalizes deviations of the pairwise similarity between datapoint representations from the similarities before fine-tuning.

The method is evaluated on clean data performance and adversarial robustness, as well as calibration error.

## Arguments for acceptance
1. The paper is well written and clear.
2. The proposed method improves on MoCo-v2 and Augmix in ImageNet accuracy, ImageNet-C corruption error, RMS calibration error and adversarial robustness.

## Arguments against acceptance
3. The proposed augmentation approach appears to be very similar to the methods used in MoCo, AugMix, and related work. Section 2.1 of the paper argues that these methods represent two different families of augmentation ($\mathcal{T}_\text{rep}$ and $\mathcal{T}_\text{rob}$). The paper suggests that augmentations need to be sampled from both of these families in a particular way (Eq. 2) to overcome the issue that one of the families affects the color histogram too much. This approach is only a small and heuristic variant of existing methods. It has been known since long before MoCo and AugMix that image augmentations improve unsupervised representation learning and robustness. Formal arguments for the distinction between $\mathcal{T}_\text{rep}$ and $\mathcal{T}_\text{rob}$ are not given. I am not sure whether the proposed variant is original enough for publication at ICLR. Further, the performance improvement of URRL over MoCov2 is small, less than one percentage point for most metrics.

4. The proposed similarity regularization leads to more significant improvements in fine-tuning performance. However, this method is not compared against baselines. Regularization approaches for improving transfer performance have been proposed before, e.g. see [Li et al., 2018](https://arxiv.org/pdf/1802.01483.pdf). These should be compared to the proposed method. In particular, well-tuned weight decay towards the frozen pre-trained weights ($L^2$-$SP$ in Li et al.) is an essential baseline. I suspect that the proposed similarity regularization may be functionally equivalent to decaying weights towards their initial values.

## Conclusion
While the paper is well organized and clear, in its current form it does not meet the originality and significance standards of ICLR and lacks crucial baselines. Suggestions for improvement:

5. Provide a more formal argument for the proposed augmentation framework to show that it is a significant conceptual advance over the current knowledge that augmentations are important.
6. Provide a more formal motivation for the similarity regularization, including a formal comparison to $L^2$-$SP$.
7. Compare the similarity regularization to a well-tuned $L^2$-$SP$-decay baseline.

## Update after rebuttal:
The authors provided convincing evidence that the proposed similarity regularization performs better with regards to robustness than $L^2$-$SP$. I adjusted my rating accordingly. I still think the contributions are borderline because the proposed regularization is not backed up by theoretical arguments and the augmentation approach is incremental.

---

> ### Author Response · Authors · 2020-11-24
> **Our Response to Reviewer 2**
>
> We thank the reviewer for suggestions to strengthen our paper.
>
> **L2-SP**: We have included L2-SP’s results in Table 2. We followed the publicly available source code released by the author. The proposed similarity regularization significantly outperforms L2-SP’s robust accuracy by almost 9% on average. The hyperparameter analysis for L2-SP on CIFAR-10 is shown in Figure 2 of Appendix A. We see that the best performing hyperparameters in [Li et al., 2018](https://arxiv.org/pdf/1802.01483.pdf) (alpha = 0.1, beta = 0.01) actually hurt both the clean accuracy and robust accuracy in this case. This is due to two key differences between our work and theirs: First, our focus is on *unsupervised* representation learning. Compared to the representations learned from supervised learning on large-scale datasets like ImageNet and Places365, our unsupervised representations typically need further updates in order to do well on downstream classification tasks. In this case, L2 regularization is not enough by itself to improve the performance. Second, *robustness* is not the goal of Li et al., 2018, and L2-SP is not designed to improve the robustness of transfer learning. In contrast, the primary goal of our work is to improve the transferable robustness of learned representations, and the design of our similarity measure loss is motivated by preserving the robustness of learned representations during fine-tuning. As discussed in Section 2.2, the robustness of a feature representation comes from how it measures the similarities between images, and the proposed approach explicitly regularizes it during fine-tuning. Our results in Table 2 also show the importance of the proposed approach.
>
> **Contribution of URRL**: We understand the reviewer’s concern that the benefits of augmentation are already known. While this is the case for supervised learning, we would like to emphasize that our work focuses on *unsupervised representation learning*, where much work remains to be done for augmentations. As noted by Reviewer 1, [recent work](https://arxiv.org/abs/2010.08377) shows that “most techniques for self-supervised learning don't improve robustness at all." In addition, recent works (e.g., [SimCLR](https://arxiv.org/abs/2002.05709) and [InfoMin](https://arxiv.org/pdf/2005.10243.pdf)) also comment on the challenge of selecting proper augmentations for unsupervised learning. In this case, “it was not obvious a priori” that our approach would work (R1). Therefore, a key contribution of our work is to address both of these challenges by improving the clean accuracy and robust accuracy *at the same time*. This upends for unsupervised learning the common assumption that there needs to be a trade-off between clean and robust accuracies ([Zhang et al., 2019](https://arxiv.org/abs/1901.08573)).
>
> **Generality of Our Approach**: We have included additional results in Table 4 of Appendix B, where we show the generality of our approach. Our approach is not limited to application to MoCo v2 studied in the main paper. Our approach (URRL) improves two other unsupervised learning approaches ([InstDis](https://openaccess.thecvf.com/content_cvpr_2018/CameraReady/0801.pdf), [PIRL](https://openaccess.thecvf.com/content_CVPR_2020/papers/Misra_Self-Supervised_Learning_of_Pretext-Invariant_Representations_CVPR_2020_paper.pdf)).

---

> > ### Comment · AnonReviewer2 · 2020-11-24
> > **Incomplete hyperparameter tuning for L2-SP baseline**
> >
> > **L2-SP**: Thank you for adding L2-SP as a baseline. Unfortunately, the hyperparameter sweep is clearly incomplete: The range of values does not include an optimum in robust accuracy for either $\alpha$ or $\beta$ (Figure 2b). In fact, the $\sim 20$-point increase of robust accuracy from $\alpha=0.01$ to $\alpha=0.001$ suggests that significant improvement can be expected by further reducing $\alpha$. In addition, clean accuracy is still significantly better for L2-SP than for SR on average (Table 2).
> >
> > It would also be good to address R1's concerns regarding SR (*"Logit pairing can lead to an overestimate of adversarial robustness. Please include results with an adversarial attack with restarts."*)
> >
> > **Contribution of URRL**: Thank you for your response and the additional results for InstDis and PIRL. I understand that your method applies to unsupervised learning. My original assessment still stands: I am not sure whether the proposed variant (of augmentation) is original enough for publication at ICLR. Further, the performance improvement of URRL over MoCo2 (and the other SSL methods) is small, less than one percentage point for most metrics. As the other reviewers state, the main novelty is in the SR regularization, making the comparison to L2-SP all the more important.

---

> > > ### Author Response · Authors · 2020-11-25
> > > **L2-SP Hyperparameter Sweep**
> > >
> > > We understand the reviewer’s concern that the hyperparameter sweep could be incomplete. We have further reduced alpha by two orders of magnitude and include results for alpha = 0.0001 and alpha = 0.00001 in Figure 2 of Appendix A. The result that is not using L2-SP is also included as a reference. We can see that by further reducing alpha, the robust accuracies converge closer to the result that is not using L2-SP. In this case, the ~20% difference between alpha = 0.01 and alpha = 0.001 is explained by the fact that alpha = 0.01 reduces robust accuracy by ~20% compared to that of “no L2-SP,” and alpha = 0.001 is performing similarly to that of “no L2-SP.” We are continuing to revise our paper to incorporate further experiments suggested by the reviewer.

---

### Official Review · AnonReviewer4 · 2020-10-30
**A new similarity loss to fine-tune representations using auxiliary tasks.**

**Rating:** 7
**Confidence:** 4

**Review:**

The authors improve the fine-tuning of a data representation allowing
resistance and recognition of adversarial or corrupt inputs.  Their method,
URRL, uses an auxiliary unsupervised task and robust data augementation to
improve performance for both clean and altered inputs.  A key addition is that,
while fine-tuning on the auxiliary (downstream) task, performance on the
original data is maintained by favoring small differences between original
robust representation and the fine-tuned representation.

The work is well presented, and the main significance is in providing a fast
way to partially gain some of the robustness available in much more difficult
adversarial training.  This might be of significance in applications like
online learning, where only fast fine-tuning is practical.

Their approach limits how much fine-tuning is allowed to deviate from the
original, robust representation.  It is a *fast* way to partially recover some
of the benefits of a much more expensive adversarial training.  The problem
they address is important.  For example, observations in related works often
note a slight decrease in performance on the original supervised task during
later fine-tuning.  The issue to resolve is how to nicely limit degradation,
even in face of aggressive or long-term fine-tuning.

To maintain robustness during fine-tuning, one (expensive) option is to
adversarially fine-tune the downstream task.  The authors propose a quicker
approach: a loss function maintaining similarity between fine-tuned and
original representation (e.g. low fine-tuning distortion of dot-product
similarity).

The experiments nicely evaluate robustness with a number of relevant metrics,
using ImageNet with many corruption types.  Results are presented as improvements
to the MoCo v2 codebase.  The use of a linear classifier stage seemed reasonable to me.

Firstly they address what sets of data augmentations are most effective for
images, using a mechanism that probabilistically switches between augmentation
types for supervised and unsupervised tasks.

Secondly, the more interesting results with similarity loss showed very good (usually best)
adversarial performance.  Accuracy on clean data was degraded by ~ 10%, still less than the
degradation caused by adversarial training.  Similarity loss can be applied to other existing
techniques (like their MoCo baseline).

Q: Does the similarity measure loss still make sense if fine-tuning uses
data sampled from a new/changing input distribution?  It might
(see also "Test-Time Training with Self-Supervision
for Generalization under Distribution Shifts", Sun et al.)

---

The authors' comments strengthen the argument for the method.   I (perhaps liberally) conceptualize
their approach as one of tethering "close to a rotation" but still don't have a simple understanding of
why/when this should be better than tethering "close to original parameters". My original rating is unchanged.

---

> ### Author Response · Authors · 2020-11-24
> **Our Response to Reviewer 4**
>
> We thank the reviewer for the helpful discussions and comments.
>
> **Applicability to Changing Input Distribution**: We believe that the proposed similarity measure loss can still be helpful in settings like [Sun et al., 2020](https://arxiv.org/abs/1909.13231). In their setting, given an unlabeled test sample, they first update the model parameters using a self-supervised loss before making a prediction. Our regularization is also unsupervised and can be directly applied to the unlabeled test sample jointly with the self-supervised loss update to regularize it. This could prevent the self-supervised loss to drastically update the trained weights with small sample sizes. Test-time regularization is a promising direction, and we have updated our paper to include a discussion of Sun et al., 2020.

---

### Author Response · Authors · 2020-11-24
**Our Response to All Reviewers**

We thank all reviewers for the constructive suggestions and comments. We appreciate that the reviewers recognize the contributions of our work and that “it could easily be broadly applicable to the burgeoning area of self-supervised learning” (Reviewer 1).

The main concerns of Reviewer 2 are: (i) Comparison to a transfer learning regularization baseline ([L2-SP](https://arxiv.org/pdf/1802.01483.pdf)), and (ii) The benefit of data augmentation is already known.

We address both of these concerns in our revised draft and responses by the following:

**L2-SP**: We include a detailed analysis and comparison with L2-SP and show that the proposed similarity regularization still significantly outperforms L2-SP’s robust accuracy by almost 9% on average.

**Contribution of Our Approach (URRL)**: We clarify that much work still remains to be done for applying data augmentations to *unsupervised representation learning*. This challenge is noted by several recent works ([SimCLR](https://arxiv.org/abs/2002.05709), [InfoMin](https://arxiv.org/pdf/2005.10243.pdf), [Geirhos et al., 2020](https://arxiv.org/abs/2010.08377)) and nicely summarized by Reviewer 1. Our work contributes by showing that it is possible to improve both the clean accuracy and robust accuracy at the same time with a simple and general approach.

**Generality of Our Approach**: In addition, we follow Reviewer 1’s advice and apply the proposed URRL to two additional unsupervised learning approaches ([InstDis](https://openaccess.thecvf.com/content_cvpr_2018/CameraReady/0801.pdf), [PIRL](https://openaccess.thecvf.com/content_CVPR_2020/papers/Misra_Self-Supervised_Learning_of_Pretext-Invariant_Representations_CVPR_2020_paper.pdf)), and URRL improves both of these methods (Appendix B). This supports the generality of our approach, and we hope this facilitates its broad adoption in self-supervised/unsupervised learning.

---

### Decision · Program_Chairs · 2021-01-07
**Final Decision**

**Decision:**

Reject

**Comment:**

The authors propose a framework which combines pretext tasks and data augmentation schemes with the goal of improving robustness of image representations. The authors show that this approach empirically can lead to increased accuracy on both corrupted and uncorrupted data simultaneously. Furthermore, the authors propose a regularization procedure which can be used to maintain a robust representation during transfer to arbitrary downstream tasks.

The studied problem is significant and highly relevant to the ICLR community. The reviewers agreed that the work is timely, and appreciated the clarity of exposition. At the same time, the reviewers remained in disagreement in terms of novelty and significance of the results -- the proposed method is seen as a clear incremental application of existing techniques in the self-supervised setting. The authors argue that it was not clear that such augmentations would improve the robustness as well as accuracy, but these methods were developed and optimised to improve robustness. In fact, in [1] the authors conclude that “...today's supervised and self-supervised training objectives end up being surprisingly similar” as well as point out that SimCLR is more robust than competing self-supervised methods. Hence, establishing that there is indeed some empirical benefit is a step in the right direction, but not sufficient to meet the bar of acceptance. Furthermore, given the recent trend of scaling-up existing approaches, in particular in terms of the neural architectures and the batch sizes, the computational costs of the proposed regulariser in Eq (4) coupled with the additional hyperparameter to optimize make the approach less practical and general than claimed. In addition, the reviewers pointed out the lack of comparison to a proper baseline, as well as the issue with hyperparameter selection for the baseline after the author response. Finally, given access to additional data augmentations and one more hyperparameter to tune the results should substantially outperform the baselines.

For the reasons outlined above and the incremental nature of the work, I will recommend rejection. That being said, this was a borderline case, and I urge the authors to carefully revise the manuscript with the received feedback.

[1] https://arxiv.org/abs/2010.08377